# Growth Rate and Bone Hydroxyproline Concentration in Turkeys Fed with a Silage-Composed Diet Modified with Different Diet Cation–Anion Differences (DCADs)

**DOI:** 10.3390/ani12010066

**Published:** 2021-12-29

**Authors:** Marta Wójcik, Klaudia Stachal, Mateusz Burzec, Kamil Gruszczyński, Agnieszka Korga-Plewko

**Affiliations:** 1Sub-Department of Pathophysiology, Department of Preclinical Veterinary Sciences, Faculty of Veterinary Medicine, University of Life Sciences in Lublin, Akademicka 12, 20-033 Lublin, Poland; stachalklaudia@gmail.com (K.S.); burzecmat@gmail.com (M.B.); kamil.gruszczynski@animal-pharma.com (K.G.); 2Independent Medical Biology Unit, Medical University of Lublin, 8b Jaczewski Street, 20-090 Lublin, Poland; agnieszkakorga@umlub.pl

**Keywords:** hydroxyproline, body weight, corn silage, diet cation–anion difference, turkey

## Abstract

**Simple Summary:**

In this study, we point to the possibility of using dietary cation–anion difference manipulation to improve growth rates and bone condition, especially in the last phase of poultry turkey husbandry, when birds have many bone abnormalities.

**Abstract:**

Our goal was to determine the responses of body weight (BW) and bone hydroxyproline (Hyp) concentration in turkeys fed a corn silage (CS) diet with different values of dietary cation–anion differences (DCADs). The turkeys (n = 90) were divided into five groups and fed as follows: group A (control)—standard diet (SD) (60%) plus CS (40%); group B—SD (60%), CS (40%) plus 240 g of CaCl_2_ per 100 kg of diet; group C—SD (60%), CS (40%) plus 480 g of CaCl_2_ per 100 kg of diet; group D—SD (60%), CS (40%) plus 240 g of NaHCO_3_ per 100 kg of diet; group E—SD (60%), CS (40%) plus 480 g NaHCO_3_ per 100 kg of diet. The addition of a lesser amount of CaCl_2_ lowered the DCAD, which ranged between 52.5 ± 4.19 and 91.14 ± 3.14 mEq/kg DM. An increased content of CaCl_2_ led to high negative values of DCAD. NaHCO_3_ supplemented in both doses resulted in a significant elevation of DCAD. Compared to each experimental group, feeding birds with a diet supplemented only with CS resulted in a lower BW. Addition of CaCl_2_ or NaHCO_3_ to the diet improved BW, but only CaCl_2_ addition enhanced the bone Hyp amount. In conclusion, we suggest that an anionic diet with low DCAD can prevent bone abnormalities in large turkeys, especially in the final course of production.

## 1. Introduction

One alternative food for broiler turkeys might be the incorporation of inexpensive corn silage to the typical formula [1]. The addition of corn silage to the daily meal is well accepted by turkeys. However, very little attention has been paid to the manipulation of the diet cation–anion difference (DCAD) in growing turkeys. Dietary sodium, potassium, chloride, and sulfate, which are referred to “strong ions”, exert a great effect on the acid–base balance of body fluids [2]. Thus, DCAD is important in blood pH regulation, for better enzymatic efficiency, and influences bird growth and performance. Because one of the reasons for the silage suppression of the growth rate may be the deviation in DCAD, we sought to determine whether acidification or alkalization of the diet may improve the growth of young turkeys. Halley et al. indicated that altering dietary cation–anion ratios influenced not only poultry growth but also caused leg abnormalities such as valgus–varus deformities and dyschondroplasia, generally reported as “bowed and crooked” legs [3]. Weak bones result in breakage during processing and a lower meat grade. Moreover, as described by Onyango et al., weak legs often result in reduced feed intake thus affecting weight gain [4]. 

According to current knowledge, collagen type I, which provides bone with tensile strength, is the major protein component in bone, comprising over 90% of the organic bone matrix [5]. The amino acid sequence of collagen is rich in proline. Collagen is formed from AAs (mostly glycine and proline) by fibroblasts through the pathway of intracellular protein synthesis, which includes AA activation, initiation of peptide formation, peptide elongation, termination, and post-translational modifications [6]. Approximately 50% of the proline sidechains are hydroxylated post-translationally to form hydroxyproline (Hyp) by the enzyme proline hydroxylase, which occurs even before the completion of the polypeptide chain synthesis. All animals generate Hyp from proline, and this metabolic process requires a large amount of energy (6 mol ATP/mol Hyp), which includes 4 mol ATP for collagen synthesis and 2 mol ATP for collagen degradation [6,7,8]. Thus, direct provision of proline and Hyp or their immediate precursors in diets will minimize the expenditure of energy for the endogenous synthesis of these two AAs. However, some authors emphasize that Hyp in the tropocollagen molecule is derived by an in situ hydroxylation of proline during the early stages of collagen biosynthesis by osteoblasts and not from dietary sources [6]. During collagen breakdown, Hyp is released from bone and not recycled to form new collagen [5,6]. Thus, serum and urinary Hyp concentration are therefore considered to be an osteoclastic bone resorption marker. 

Although collagen is the major organic component of bone matrix, there are few studies on the effect of DCAD on collagen synthesis in bone. Our approach intended to determine the responses of turkeys to a corn silage diet with different values of DCAD in respect to their growth and to clarify the response of bone Hyp concentration in turkeys fed with a corn silage diet with different DCADs.

## 2. Materials and Methods

### 2.1. Birds Husbandry

Ninety 28 day old female turkeys (BUT-6) were used in our experiment, which was carried out in floor pens (2.5 × 2.5 m^2^) arranged by blocks at the turkeys’ farm in Petryłów, Poland. At the beginning of the experiment, the birds were weighed and divided into 5 groups with 18 birds per pen/group. The birds were housed in an environmentally controlled room according to standard turkey management practice. From the 4th week of the birds’ age and to the 14th week of age, birds were provided with a 3-phase feeding program: grower I was conducted for 4 weeks (22–49 days of age), grower II for 4 weeks (50–77 days of age), and finisher for 2 weeks (78–91 days of age). All experimental procedures were approved by the Local Ethics Committee of Animal Care at the University of Life Sciences in Lublin (No. 18/2014). 

### 2.2. Diets

The turkeys in each group were fed as follows: group A (n = 18) (control)—standard diet (60%) plus corn silage (40%); group B (n = 18)—standard diet (60%), corn silage (40%) plus 240 g of CaCl_2_ per 100 kg of diet; group C (n = 18)—standard diet (60%), corn silage (40%) plus 480 g of CaCl_2_ per 100 kg of diet; group D (n = 18)—standard diet (60%), corn silage (40%) plus 240 g of NaHCO_3_ per 100 kg of diet; group E (n = 18)—standard diet (60%), corn silage (40%) plus 480 g NaHCO_3_ per 100 kg of diet. In each group, the minced corn silage was mixed with a standard diet, giving a homogenous mass. The composition of standard diet and corn silage is given in Table 1 and Table 2. Feed was offered ad libitum, and drinking water was continuously supplied. 

### 2.3. Performance Measurements

Individual bird weight was measured weekly to estimate average body weight (BW).

### 2.4. Analytical Procedures

To obtain a constant dry weight of diet samples, each of them was dried at 105 °C (Muffle Furnace FCF 22SP, Alchem Group Ltd., Totuń, Poland). Then, for the analysis of diet ion contents, 0.5 g samples weighted at a ±0.0001 accuracy were digested in a Multiwave 3000, Anton Paar microwave stove (Anton Paar Poland, Warsaw, Poland). Eighteen microliters of digested mixture consisting of 70% perchloric acid and 60% nitric acid (5:1, *v*/*v*) were added to 0.5 g of every sample. After a double-phase mineralization process was conducted: phase I—180 °C/20 min; phase II—220 °C/90 min. The samples were then transferred to a calibrated tube, and the volume was increased to 25 mL with deionized water. From each kind of diet, 3 samples were analyzed [9].

### 2.5. Diet Ion Contents

The DCAD values were analyzed as described previously [1,10]. In brief, for the analysis of diet Na, Ca, K, Mg, and S contents, an ICP-OES (inductively coupled plasma optical emission spectrometer) equipped with a charge-injection device (CID) detector was used. Control of the spectrometer was provided by the PC-based iTEVA software. The following parameters were used: RF generator power of 1150 W, RF generator frequency of 27.12 Mhz, coolant gas flow rate of 16 L/min, carrier gas flow rate of 0.65 L/min, auxiliary gas at 0.4 L/min, max integration times of 15 s, pump rate of 50 rpm, axial viewing configuration, 3 replicates, and flush time of 20 s. The multi-element stock solution (Inorganic Ventures) contained: ^40^Ca, ^30^K, ^24^Mg, ^23^Na, ^32^S, and ^31^P in 2% HNO_3_—1000.00 mg/L (ppm) (Analityk, Warsaw, Poland).

The measurements of the chloride content in the diet were performed using the direct potentiometric method with a liquid membrane for a selective chloride ion electrode. Samples of 0.5 mL were mixed with 4.5 mL of deionized water, and 100 µL of ISA (ionic strength adjuster—5 M NaNO_3_), was added to adjust the ionic strength of the samples. Two standard chloride solutions with concentrations of 5 mg·L^−1^ and 10 mg·L^−1^ were used to calibrate the measuring device (Orion 920A digital ion-analyzer; Orion Research Inc., Boston, MA, USA). The diet cation–anion difference was calculated from DCAD (mEq/kgDM) = (Na + Ca + Mg + K) − (S + Cl) [10].

### 2.6. Hydrolysis of Femur Samples

At the end of the experiment, all the birds in each group were humanely killed by cervical dislocation to collect bones for the determination of hydroxyproline. Turkey femurs were cleaned of all soft tissue and were cut into 0.5–1 cm strips and dried to a constant weight at 100 °C. Dry bone particles were ground through a bone grinder, and homogenous samples were saved for analyses. Hydroxyproline concentration was measured by the method of Monnier et al. as a marker for collagen content [11]. In brief, 40 mg of powdered bone samples, in duplicate, were delipidated overnight in a chloroform-methanol solution (2:1, *v*/*v*). After rehydration, hydrolyzation and evaporation samples were reconstituted in 250 µL distilled H_2_O and filtered with a centrifuge tube filter. Hyp concentrations were analyzed at 564 nm with a Cecil CE2021 spectrophotometer.

### 2.7. Blood Analysis

Before the experiment and at the 6th, 8th, 10th,12th, and 14th week, venous blood samples were collected into heparinized (50 IU/mL^−1^) Monovette syringes via a puncture of the wing vein (i.e., branchial vein). Blood was drawn directly from the syringes into a blood gas/electrolyte analyzer (ABL80 Flex, Radiometer, Copenhagen, Denmark) for an immediate analysis of the pH value. The pH values were corrected to reflect a body temperature of 41.5 °C [1].

### 2.8. Statistical Analysis

The statistical analysis of the obtained results was carried out using the software Statistica 13.0 (StatSoft, Kraków, Poland). All results are presented as the mean values with standard deviations (mean ± SD). The normality of data distribution was tested using the Shapiro–Wilk test. The Hyp bone concentration and blood pH results were tested using one-way ANOVA analysis of variance (diet supplementation) and Tukey’s post hoc test. Tukey’s test was used to compare differences between means when the results were declared significant (*p* < 0.05). Other results (DCAD, BW) were analyzed by Student’s *t*-tests. For all tests, the criterion of an α level of *p* ≤ 0.05 was used to determine statistical significance. The Pearson correlation coefficients were calculated using the Statistica 13 software to investigate the possible influence of DCAD values on both BW and Hyp bone concentration. The coefficient *r* indicates the degree of linear correlation between the two variables.

## 3. Results

The values of the control and experimental DCAD obtained in grower I, grower II, and finisher rations are summarized in Table 3. Under control conditions, all DCAD values were positive and reached the maximum (141.66 ± 8.23 mEq/kg DM) in grower II. The addition of the lesser amount of CaCl_2_ to the diet significantly (*p* ≤ 0.05) lowered the DCAD, ranging between 52.5 ± 4.19 and 91.16 ± 3.14 mEq/kg DM in grower I and finisher, respectively. An increased content of CaCl_2_ led to high negative values of DCAD in each diet used in the experiment. The NaHCO_3_ supplemented in both the 240 and 480 g/100 kg diets resulted in a significant elevation of DCAD independent of the kind of diet, with the highest 332.62 ± 15.94 mEq/ kg DM in the finisher supplemented with 480 g NaHCO_3_/100 kg.

The birds fed with a standard diet and corn silage showed a lower body weight in comparison to the other groups. This lower BW was maintained up to the 11th week of growth. During the next 3 weeks of fattening, BW in this group was comparable with the BWs in group C, D, and E. However, during the last week of the experiment, a marked (*p* ≤ 0.05) decrease in BW was noticed in comparison to all other groups of birds (Figure 1 and Table 4). Compared to group A, the addition of anionic salt (CaCl_2_) in its lower dose 240 g/100 kg diet led to a significant increase in BW, which continuously increased from the 9th to the 11th week of growth. During the 12th and 13th weeks of fattening, BWs in this group was also higher than BWs in group A but without statistical significance. As shown in Figure 2, in this experimental group, BW was positively correlated with DCAD, and values reached a high correlation coefficient of r = 0.84.

The highest value of BW, averaged at 7.8 ± 0.81 kg/bird, under supplementation of NaHCO_3_ in a 240 g/100 kg diet was noticed in the 14th week of fattening. In groups supplemented with NaHCO_3_, BW poorly correlated with DCAD values. The correlation coefficient did not exceed 0.52.

In birds that consumed the standard formula and only CS (i.e., group A), the bone concentration of Hyp averaged 12.52 ± 0.69 µg/mg DM (Figure 3). The exposure of turkeys to low doses of CaCl_2_ resulted in gentle but statistically significant (*p* ≤ 0.05) increases in Hyp to values of 13.93 ± 3.07 µg/mg DM. In group C, the augmentation of Hyp was also noticed (i.e., 12.58 ± 0.84 µg/mg DM), however, without statistical significance. It should be noted that in both these experimental groups, the DCAD of the diet administrated during the last two weeks of fattening positively correlated with bone Hyp concentration (i.e., *r* = 0.95 and *r* = 0.97 in groups B and C, respectively).

Conversely, the addition of NaHCO_3_ to the diet, independent of its dose, led to visible and insignificant decreases in Hyp concentrations of 11.39 ± 0.25 µg/mg DM in group D and 10.54 ± 1.43 µg/mg DM in group E in comparison to group A. On the other hand, if these values were compared to the groups of turkeys supplemented with CaCl_2_ (i.e., groups B and C), the decrease in Hyp concentrations were statistically significant. Addition of cationic salt to the diet resulted in low and average correlations between the DCADs of diets administrated during the last two weeks of fattening and bone Hyp concentrations (*r* = 0.43 and *r* = 0.6 in groups D and E, respectively).

## 4. Discussion

Under our experimental conditions, the shifting in DCAD ratios were combined with improvements in feed intake (data not published) and body weight. Many authors indicated that such stimulatory responses occur because of alterations in gastrointestinal digestion, acid–base disequilibrium, and tissue responses to metabolic hormones [12,13,14].

Although both salts considerably enhanced appetite, the largest growth was noted in the group subjected to ratios enriched with anionic salts (CaCl_2_), both in lower and higher doses. Therefore, it is no surprise that some authors called these salt food intake regulators [15]. On the other hand, stimulatory responses of anionic salts in our turkeys’ BWs contrast with the data from broiler chicks provided by Kim et al. [16]. However, as it was mentioned in our previous study, the possibility that chloride can reduce feed intake refers only to a state of abnormal lysine levels when chloride violates the tissue patterns of basic amino acids, resulting in a decreased appetite [1]. Moreover Adedokun et al. showed that birds fed with a diet containing a high level of DCAD consumed less feed than birds on a low DCAD diet [17].

In all of our experimental groups, the mean body weight continuously increased during the 10 weeks of the production period, but it was lowest in the group subjected only to meals with CS. The growing responses to CS ratios supplemented with acidifying and alkalizing salts were most visible in the last week of the experiment with the nutrition of the finisher diet. What is interesting is that the anionic salts suppressed BW during this period, while cationic salts enhanced it under the same experimental conditions. A slightly reduced BW after consumption of anionic salts may be connected with higher levels of chloride and finally result in an impairment in weight gain and the efficiency of feed utilization [13,18,19].

According to Johnson at al., the optimal range of DCAD for birds should be between 180–300 mEq/kg DM [20]. Our results show that an increase in growth parameters is induced with a DCAD in a range between −115.20 ± 34.58 and +332.62 ± 15.94 mEq/kg DM. Positive growth response under such variations in DCAD may, as mentioned above, be associated with improved nutrient availability and their utilization.

It should be emphasized that the acidogenity or alkalogenity of a diet also influences the bone concentration of hydroxyproline. Although the acidogenic component of a diet increases Hyp in bone and NaHCO_3_ diminishes the bone concentration of this amino acid. Moreover, the amount of Hyp in bone in the group supplemented with cationic salt was lower than control group fed with a standard diet and CS.

Hydroxyproline as well as its precursor proline are unique amino acids (AAs), both chemically and biochemically. Moreover, remarkable differences in proline and hydroxyproline metabolism among species was observed [6]. In contrast to mammals, birds have low arginase activity in tissues and, therefore, a limited ability to convert arginine into proline. Thus, birds have a physiological deficiency of substrate to hydroxyproline production. One way to solve this problem is dietary supplementation. Several lines of experimental evidence indicate the essential requirement for proline as a nutrient for poultry. For example, first, supplementing 0.0, 0.2, 0.4, and 0.8% proline to a chemically purified diet containing 1% arginine and 10% glutamate dose dependently increased daily weight gains (from 11.88 to 13.38 g/day) in young chickens without affecting their feed intake (for an average of 114 g/chick) [6]. Until recently, hydroxyproline has been traditionally considered to have little nutritional significance, but now it is recognized as a substrate for the synthesis of glycine as an essential AA for chickens and turkey. Additionally, in ruminants, Hyp acts as a substrate for synthesis of pyruvate and glucose. This AA may also scavenge oxidants and regulate the redox state of cells. Furthermore, hydroxyproline may greatly impact the nutrition of birds, which cannot sufficiently synthesize glycine from other AAs.

Based on our previous studies, blood pH under a prolonged acidogenic diet is in the range between 7.36 and 7.38, which is low but still a physiological value, and it is called “chronic metabolic acidosis” (1). Many authors, including Kraut J.A. and coworkers, indicate that metabolic acidosis has been implicated in the pathogenesis of both osteomalacia and osteopenia thus influencing bone formation and bone resorption [21]. However, most of the current evidence relates to skeletal and plasma calcium balance and bone mineralization processes rather than bone hydroxyproline levels. Several separate results have attempted to explain the inhibitory effect of acidosis on bone mineralization. Brando-Burch A. et al. observed that osteoblast ALP activity was strongly reduced at low pH [22]. This enzyme is thought to participate in the mineralization process by cleaving phosphate groups from organic compounds, thereby increasing local concentrations of inorganic phosphate. The same authors suggest that for optimal mineralized bone formation to occur in vivo, extracellular pH in the immediate osteoblast environment should be ≥7.20. Given the other data, plasma levels of amino acids, such as threonine, serine, asparagine, citrulline, valine, leucine, ornithine, lysine, histidine, arginine, and also hydroxyproline, increased significantly with the induction of acidosis [23]. These results confirm that acidosis in humans is a catabolic factor stimulating protein degradation and amino acid oxidation. On the other hand, Bushinsky and colleagues have reported that in mouse osteoblasts, metabolic acidosis also inhibits the production of collagen and other matrix components [24]. In turn, Ramp and co-workers found that glycolysis, alkaline phosphatase activity, and collagen production by primary chick and human osteoblasts was reduced at low pH, and they hypothesized that osteoblast function may be optimal at pH 7.2 [25]. It should be emphasized that in our experiment supplementation of turkey with an acidogenic diet with low DCAD values never resulted in a pH lower than 7.2 (Table 5). Moreover, according to the modern application of Stewart’s physicochemical approach to acid–base balance (ABB), variations in individual components alter plasma pH minimally. Thus, if Cl^−^ anions increased the following consumption of CaCl_2_, the supplementation of Ca^2+^ does not change the ABB. It is thus justified to assume that these supplements supported not only the growth responses of experimental turkeys but also enhanced their bone Hyp concentration. What is important is that the further enhancement of CaCl_2_ content in the diet (i.e., group C) did not correlate with bone Hyp level, which started to decrease. This means that only gentle manipulation of the dietary cation and anion amounts leads to Hyp bone content improvement.

## 5. Conclusions

In conclusion, the changes in DCAD during the feeding of diet containing CS and anionic salts exerted stimulatory effects on turkeys’ body weight. However, this stimulation of BW correlated with high Hyp bone concentration only when broiler turkeys were fed with CS diet supplemented with a low dose of CaCl_2_. If so, we can assume that an anionic diet with low DCAD might be used to prevent bone abnormalities in large turkeys, especially in the final course of production. However, to confirm this hypothesis, additional analyses, such as histomorphometric analysis and a three-point bending test of bones, must be performed in future experiments.

## Figures and Tables

**Figure 1 animals-12-00066-f001:**
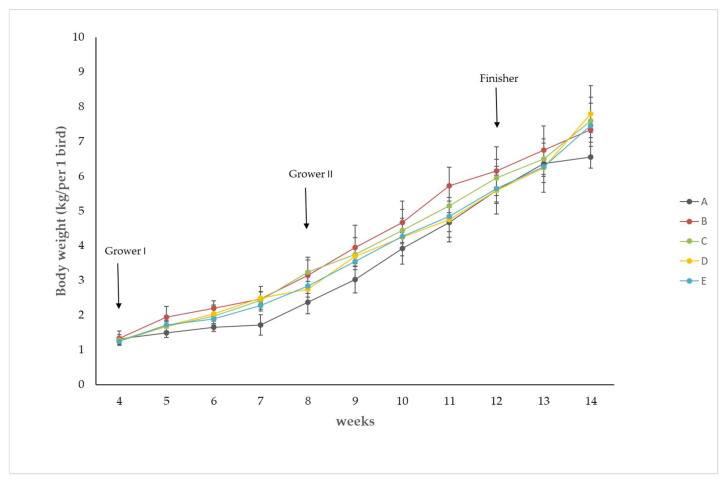
Time course of body weight change as influenced by corn silage and different diet cation–anion differences. To better visualize the statistical significance, values from Figure 1 are additionally presented in Table 4.

**Figure 2 animals-12-00066-f002:**
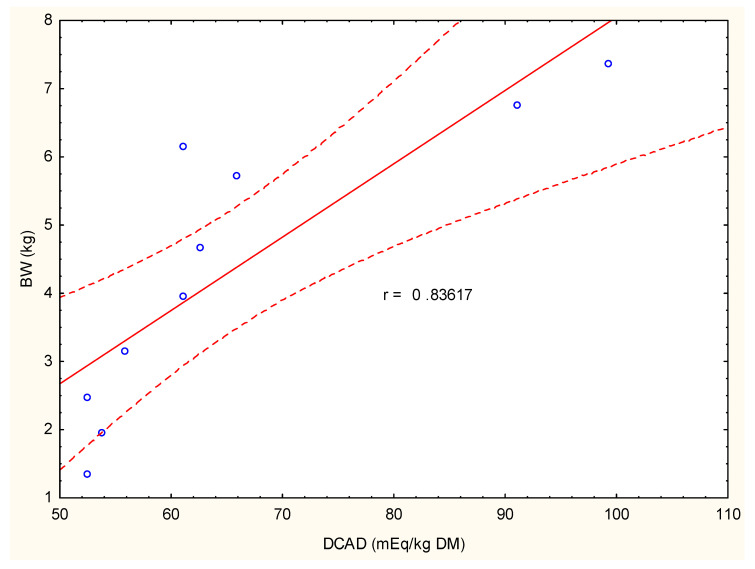
High positive correlation (r = 0.830) between BW and DCAD value in the group of birds fed the standard diet (60%), corn silage (40%) plus 240 g of CaCl_2_ per 100 kg of diet (i.e., group B). Estimation was performed with Statistica 13 (StatSoft, Poland) using linear Pearson matrix correlation analysis.

**Figure 3 animals-12-00066-f003:**
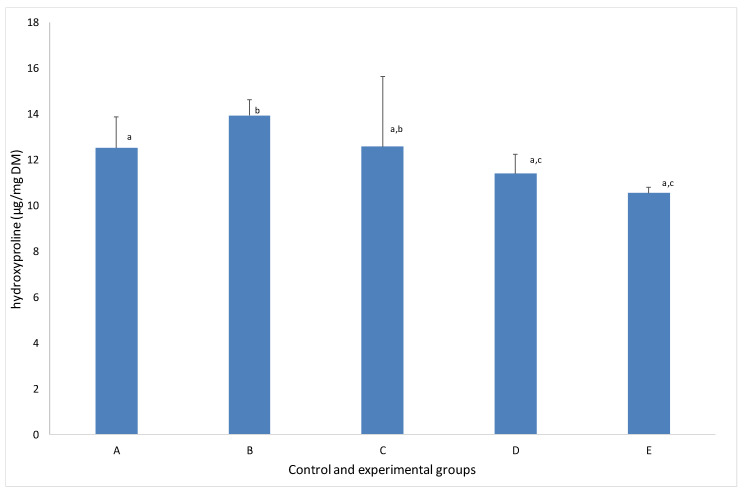
Bone hydroxyproline concentration in the control and experimental groups. Values are the mean ± SD. (a, b, c) Values in rows with different letters differ significantly (*p* ≤ 0.05; Tukey’s HSD test).

**Table 1 animals-12-00066-t001:** Ingredients and composition of the standard diet and corn silage for turkey broilers according to the manufacturer’s specification (De Heus, Animal Feed Industry, Łęczyca, Poland). Daily rations averaged 2 kg of grower I plus CS, 5 kg of grower II plus CS, and 5 kg of finisher plus CS.

Ingredients (%)	Grower I	Grower II	Finisher
Wheat	29.208	22.045	23.605
Soybean meal	27	19.567	13.3
Yellow corn	20	25	25
Triticale	15	25	30
Soybean oil	3.1	3.233	3.233
Hemoglobin	1.5	1.5	1.5
Fodder chalk	1.433	1.167	0.967
1-Ca Phosphate	0.933	0.667	0.517
PRMX BR IND 0.5% *	0.5	0.5	0.5
L-Lysine HCl	0.277	0.267	0.297
L-Methionine	0.243	0.203	0.197
NaHCO_3_	0.2	0.2	0.2
NaCl	0.147	0.15	0.153
Toxi-Tect **	0.1	0.1	0.1
Noack AC BIL 1 Na ***	0.1	0.1	0.1
Choline chloride	0.087	0.077	0.07
L-Treonina	0.067	0.083	0.077
Arg90/Val10 ****	0.06	0.097	0.15
Avizyme	0.025	0.025	0.025
Axtra PHY, g/kg	0.02	0.02	0.01
Nutrients			
MEpoultry, kcal/kg	3025.24	3125.25	3199.47
MEpoultry, MJ/kg	12.66	13.08	13.39
Humidity, g/kg	119.51	119.27	119.51
Crude protein, g/kg	215.29	187	167.01
Crude fat, g/kg	50.41	52.6	52.24
Crude ash, g/kg	55.99	48.21	42.1
Crude fiber, g/kg	28.02	26.26	25.59
P-inositol, g/kg	2.4	2.25	2.12
Calcium, g/kg	10.86	9.5	8.04
Phosphorus, g/kg	5.87	5.02	4.49
AvPhosph poultry, g/kg	5.41	4.73	4.02
Phosphorus Abs poultry, g/kg	4.86	4.23	3.6
Sodium, g/kg	1.81	1.81	1.71
Chlorine, g/kg	2.34	2.4	2.55
Lysine, g/kg	12.99	11.07	9.81
Lysine dig poultry, g/kg	11.61	9.89	8.78
Methionine, g/kg	5.45	4.71	4.39
Met dig poultry g/kg	5.14	4.45	4.15
Meth + cyst, g/kg	8.95	7.86	7.29
Meth + cyst dig poultry, g/kg	8	7	6.51
Threonine, g/kg	8.26	7.31	6.35
Threonine dig poultry, g/kg	7	6.22	5.4
Tryptophane, g/kg	2.64	2.22	1.92
Isoleucine, g/kg	8.23	6.86	5.81
Arginine, g/kg	13.44	11.52	10.19
Arginine dig poultry, g/kg	12.1	10.3	0.07
Choline added, mg/kg	564.2	499.1	455.7
Choline chloride added, mg/kg	650	575	525
Dry matter, g/kg	880.49	880.73	880.49
Sodium analitic, g/kg	1.4	1.4	1.41
Calcium analitic, g/kg	8.87	7.51	6.36
6-Phyt EC 3.1.3.26, FTU/kg	1000	1000	500
Protease EC 3.4.21.62, U/kg	4000	4000	4000
Endo-1,4-betaxynalase, U/kg	2300	2300	2300
Alfa-amylase EC 3.2.1.1, U/kg	400	400	400
Lasalocid sodium, mg/kg	90	--------	--------

* Lasalocide A sodium salt: 20,000.00 mg/kg; niacin: 12,000.00 mg/kg; folic acid: 400.00 mg/kg; vitamin DL: 200,000.00 IU/kg; vitamin D3: 800,000.00 IU/kg; vitamin A: 200,000.00 IU/kg, vitamin E: 15,000.00 mg/kg; vitamin K3: 600.00 mg/kg; vitamin B1: 400.00 mg/kg; vitamin B6: 1000.00 mg/kg; calcium D-pantothenate: 3913.00 mg/kg; biotin: 50,000.00 µg/kg; vitamin B12: 4000.00 µg/kg; vitamin B2: 1600.00 mg/kg; iron (II) sulphate: 8000.00 mg/kg; selenium: 60.00 mg/kg; antioxidants: propyl gallate: 7.00 mg/kg; butylhydroxytoluene (BHT): 82.00 mg/kg; De Heus, Łęczyca, Poland. ** Bentonite–Montmorillonite (E 558), Saccharomyces cerevisiae yeast, and vitamin E (dl-alpha-tocopherol acetate adsorbed on silica, minimum 50% dl-alpha-tocopherol acetate), Wipasz, Poland. *** Feed additive based on formic, propionic, and citric acid. Noack Polen Ltd., Poland. **** Arginine 90% and Valine 10%, CJ Europe GmbH—part of CJ BIO, Warsaw, Poland.

**Table 2 animals-12-00066-t002:** Chemical composition of ingredients corn silage.

DM	27%
CP% of DM	8.6
NDF% of DM	43.0
ADF% of DM	26.1
Lignin% of DM	4.0
Ether extract% of DM	2.9
Ash% of DM	3.0
Calcium% of DM	0.33
Phosphorous% of DM	0.28

DM = dry matter; CP = crude protein; NDF = neutral-detergent insoluble fiber; ADF = acid-detergent insoluble fiber.

**Table 3 animals-12-00066-t003:** Values of diet cation–anion differences obtained in the control and experimental turkeys.

Control and Experimental Groups	Grower I, mEq/kg DM	Grower II, mEq/kg DM	Finisher, mEq/kg DM
A	127.30 ± 21.3	141.66 ± 8.23	139.3 ± 18.19
B	52.5 ± 4.19 *	61.10 ± 12.37 *	91.16 ± 3.14 *
C	−100.56 ± 7.18 *	−101.35 ± 17.21 *	−115.20 ± 34.58 *
D	255.38 ± 39.76 *	242.67 ± 4.18	214.02 ± 19.13 *
E	315.68 ± 44. 6 *	310.10 ± 29.14 *	332.62 ± 15.9 *

Values are the mean ± SD obtained from replicates of the determining ions used for the DCAD calculation. * Significant differences at *p* ≤ 0.05 vs. DCAD values obtained in corn silage-fed turkeys (group A) (student’s *t*-tests).

**Table 4 animals-12-00066-t004:** Values are the mean ± SD. * Significance at *p* ≤ 0.05 vs. the BW results in group A (student’s *t*-tests).

	4th	5th	6th	7th	8th	9th	10th	11th	12th	13th	14th
A	1.32	1.5	1.65	1.72	2.38	3.03	3.93	4.67	5.6	6.38	6.55
	±0.13	±0.14	±0.11	±0.29	±0.33	±0.38	±0.45	±0.43	±0.37	±0.39	±0.31
B	1.34	1.95 *	2.2 *	2.48 *	3.15 *	3.95 *	4.68 *	5.73 *	6.15	6.75	7.35 *
	±0.21	±0.31	±0.21	±0.36	±0.53	±0.64	±0.62	±0.54	±0.70	±0.70	±0.76
C	1.25	1.68	1.98	2.43	3.25 *	3.75	4.45	5.15	5.95	6.5	7.6 *
	±0.13	±0.24	±0.31	±0.26	±0.34	±0.48	±0.60	±0.62	±0.54	±0.57	±0.68
D	1.28	1.7	2.04	2.49 *	2.75	3.7	4.25	4.75	5.6	6.25 ±0.70	7.8 *
	±0.08	±0.15	±0.21	±0.17	±0.22	±0.22	±0.54	±0.64	±0.69		±0.82
E	1.26	1.73	1.89	2.28	2.85	3.55	4.28	4.85	5.65	6.28	7.46 *
	±0.08	±0.10	±0.17	±0.10	±0.13	±0.13	±0.17	±0.44	±0.39	±0.45	±0.35

**Table 5 animals-12-00066-t005:** Influence of different DCAD diets on blood pH values in turkeys (n = 18; x ± SD). * Significantly differences at *p* ≤ 0.05 (Student’s *t*-test) vs. pH values obtained before experiment.

Control and Experimental Groups	Blood Collection Period					
	Before Experiment	6th Week	8th Week	10th Week	12th Week	14th Week
A	7.36	7.37	7.37	7.36	7.37	7.37
	±0.24	± 0.55	±0.2	±0.2	±0.2	±0.96
B	7.37	7.34 *^,a^	7.35 *^,a^	7.35 *^,a^	7.36	7.35
	±0.75	±0.77	±0.3	±0.1	±0.05	±1.04
C	7.37	7.34 *^,a^	7.34 *^,a^	7.34 *^,a^	7.34 *^,a^	7.35
	±0.02	±0.55	±0.21	±0.21	±0.21	±0.63
D	7.37	7.38	7.39 *^,a^	7.38 *	7.39 *^,a^	7.40 *^,a^
	±0.98	±0.49	±0.15	±0.15	±0.15	±0.40 *
E	7.36	7.39 ^a^	7.40 *^,a^	7.42 *^,a^	7.42 *^,a^	7.44 *^,a^
	±0.50	±0.04	±0.26	±0.6	±0.8	±0.69

^a^ values in rows with different letters differ significantly within the group (*p* ≤ 0.05; Tukey’s HSD test).

## Data Availability

The data presented in this study are available on request from the corresponding author.

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
