# Peer review of "Growth Rate and Bone Hydroxyproline Concentration in Turkeys Fed with a Silage-Composed Diet Modified with Different Diet Cation–Anion Differences (DCADs)"

_animals, 2021, doi:10.3390/ani12010066_

Round 1

Reviewer 1 Report

Thank you for this interesting approach to bone disorders in turkeys. Please see my comments and suggestions in the file attached.

Author Response

Thank you once again for revision of our manuscript. Please see the attachment

Reviewer 2 Report

My concerns have been responded.  The manuscript can be accepted after minor revision. 

1. Line 28-29. "Compared to each experimental group, the addition of CS to the SD resulted in a lower BW." This sentence did not reworded as stated.

Author Response

Once again thank you for your revision of our manuscript. Please see the attachment.

Reviewer 3 Report

 MAJOR CONCERNS

 _The Turkeys were fed in three phases; grower I, grower II, and finisher. What age did each feeding phase start and end? Only the phase duration was stated.- The information given in Materials and Methods chapter, according to the duration of experiment and the feeding time of Grower I Grower II and Finisher, indirectly pointed to the start and the end of feeding each of these diet.

The indirect reference is what this initial concern was about.  Please state the exact day of age each diet transitioned.

 _The analytical procedure described in line 101-106 is not clear. What is the rationale for these analyses? What were the diets being analyzed for in this section?- Samples of the diets with which the turkeys were finally fed were taken to the analysis. Each diet was analysed 3 times in each feeding period of the turkeys.

Again, this section is unclear.   How were the samples analyzed?   What instrumentation were used?   What nutrients were analyzed?   Details are critical in methodology so that experiments can be replicated by others.

 _On page 4, line 147, under control conditions in which the turkeys were only fed with standard diet and corn silage had the maximum value of DCAD. Why this? - To check this situation I analysed obtained results once again. In Grover II period of fattening, the standard diet supplied with the corn silage had a lower content of chloride and sulphur anions, in comparison to other periods of fattening. This finally lead to high DCAD value.

In this case, this group is not actually a control group.

 _The result section only showed values and comparisons was made without stating the significant difference values (p values, F values, DF). Also, the result stated that there was an increase in the body weight of turkey with both CaCl2 and NaHCO3, and increased hydroxyproline concentration whereas figure 1 and 2 show no significant differences.- At some points it seems that the BW values differ significantly, but the relatively high SD causes no statistical significance.

The results are still not clear.   The methods in lines 141-150 state data are analyzed by ANOVA, or student’s t-test, or Pearson’s correlation.   This is very confusing.  How were the ad hoc and post hoc analyses actually done?   The results appear to only describe a subjective comparison of correlations.  The p values presented appear to only refer to the correlations.  In lines 162 – 171 the authors discuss higher and lower BW with different treatments but present no statistical data from their ANOVA to support such assertions.  Further, the authors state above that there were no statistical differences, therefore describing higher or lower body weights is inappropriate. 

Fig.2 was corrected.

 _Page 8, line 240-243, the sentence is incomplete.- I guess everything is Ok with this sentence.

This sentence, now lines 251-253 is grammatically incorrect and still incomplete sentence grammatically.  It needs to be rewritten for clarity.

 _Page 9, 258-259, the author stated that the DCAD values never resulted in pH lower than 7.2. Where are the values for the pH measured? A pH of 7.2 is very low physiologically. The authors should have also monitored respiratory rates to determine if the birds were in metabolic acidosis and therefore under metabolic stress.- In my opinion, this sentence cannot be taken out of the context of the previous and next sentences. The information remains clear if you read these 3 sentences together. The information that we did not observed a pH lower than 7.2 is only a polemic with the research obtained by Ramp et al. It is known that manipulations of DCAD lead only to minimal deviations in blood plasma pH without metabolic acidosis, what is actually indicated in the next sentence.

This reviewer did not suggest the sentence be removed.   This part of the discussion is used to relate blood pH to DCAD values and the current set of results.  However, nowhere is the measurement of blood pH shown in the methods or results.  It is great that acid-base balances were done previously by this lab, but that is not relevant to this manuscript or study.  You cannot assume the pH changes are the same across different experiments in time.   If the authors are going to relate blood pH to explain their own data, then they need to have measured pH.

By the way in our previous experiments we analysed acid–base parameters of blood plasma in turkey poultry and we observed only slight deviations of them under influence of cationic and anionic salt supplementation.

  1. The conclusion is not justified by the data. There are no mention in results or on figures of statistical differences in BW or hydroxyproline values. – Figure 2 were corrected. Also the results chapter were corrected and filled with results correlation.

Again, this is now very confusing.   The authors state above that, “At some points it seems that the BW values differ significantly, but the relatively high SD causes no statistical significance.”  The description in the Results section do not match what is shown in Figure 1. 

 _The inference that changes in hydroxyproline directly relate to bone strength assumes quite a lot. The authors only did the hydroxyproline measures on one bone, the opposite bone should have been used for 3-point breaking test, or ashing, or other standard measures of bone strength in order to validate the hydroxyproline conclusions.- You are absolutely right. My conclusions go too far. It has been just corrected. I realize that the performance of “biomechanical” analysis of bones would improve the quality of the obtained results, but I had not such possibilities during experiment.

 _In line 155, the authors state, “The addition of corn silage to the standard diet leads to a lower body weight in compari-155 son to the other groups.” However, according to their Methods EVERY group was given corn sileage thus no comparisons are possible, and this concoulsion is not appropriate.-

I don't compare turkeys that received corn silage to those that did not. The group fed with SD plus CS is compared to the groups which additionally received acidifying or alkalizing salts. Compared to those groups turkeys receiving SD plus CS had lower body weight. However, in the corrected manuscript, this sentence will be reworded so as not to be in doubt.

Then, what are the controls?  The experimental design is now more confusing. 

MINOR CONCERNS

 _Page 1, line 23, 29 and 30, abbreviated words should be written in full.- I don’t agree. In my opinion expanding abbreviations on each of these lines will render the sentence illegible.

It is standard practice and required that all abbreviations defined upon their first use.  The authors do this for abbreviations like CS and DCAD, but not for others like DM.  And in this abstracts he authors added (N = 90) but do not define if that is for each of the 5 groups, or are their n = 18 per group giving a total of 90 animals?

 _Page 2, line 45, incomplete sentence. - Everything seems fine with this sentence.

 _Use of both indented and non-indented paragraphs.

 _Subtitles are written in lowercase instead of sentence case. all these errors result from copying the text to the template prepared by the journal. Of course they have been corrected.

 _Uneven spacing between each subtitle.

There are still formatting issues.  I have published in Animals and not had these issues once manuscript went out to reviewers.   This is the purpose of the pdf review prior to final submission. 

 _Page 3, line 129, “famous” does not fit the context.- Of course, should be “femur”. It was automatically wrong corrected.

 _Incorrect reference citation.- Reference citation was corrected.

Author Response

Thank you for your revision and valuable comments. Please see the attachment.

Round 2

Reviewer 1 Report

Thank you for the effort in improving the nutritional composition of tables. Please check the attached file for minor corrections.

Author Response

Thank you once again for detailed review. Correction was done according to included comments. Please see the attachment

This manuscript is a resubmission of an earlier submission. The following is a list of the peer review reports and author responses from that submission.

Round 1

Reviewer 1 Report

Thank you for this interesting approach to bone disorders in turkeys. Please see my comments and suggestions in the file attached.

Author Response

Thank you very much for your valuable comments. My answers are under your comments, but the corrections are made in the file animals -1184898-corrected.

Best regards

Reviewer 2 Report

This research was carried out to find responses of body weight (BW) and bone hydroxyproline (Hyp) concentration in turkeys fed with a corn silage (CS) diet with different values of dietary cation-anion difference (DCAD). The authors found that diet supplementation with CaCl2 or NaHCO3 improved Body weight, but only CaCl2 addition improved bone Hyp amount. This research was interesting and can be accepted after revision.

  1. Line 28-29. The authors stated that “Compared to each experimental group, the addition of CS to the SD resulted in a lower BW. ”According to the five groups, all of the five groups added the CS.
  2. Authors stated that 108 female turkeys were selected and divided into 6 groups, with 18 birds per pen/group. While according to the results,  five groups (A-E) only contained 90 turkey. Besides, whether the body weights of the five groups had significant differences should be stated.
  3. How many turkeys in each group were selected to analyzed the Hydroxyproline concentration should be stated.
  4. Line 139. “Statistical analysis:”was repeated and should be deleted.
  5. Why the authors only compare the difference between the control and each of experimental group? Not using multiply comparison.
  6. Line 155-156. Authors stated that “The addition of corn silage to the standard diet leads to a lower body weight in comparison to the other groups. This lower BW was maintained up to the 14 th week of growing.”This description was not accurate, because all of the five groups added the corn silage.
  7. Line 159-160. The authors stated that “The highest value of BW, averaged 7.8 ± 0.81 kg/bird, undercondition of NaHCO 3 in 240g/100 kg diet was noticed. ” At which week should be added.
  8. Line 168. “2. Figures, Tables and Schemes”should not appear as a subheading.
  9. Figure 2. Whether there were significant difference between the five groups?

Author Response

Thank you very much for your valuable comments. I tried to substantively relate to your remarks. My answer are in red color.

Best regards

Reviewer 3 Report

Growth Rate and Bone Hydroxyproline Concentration in Turkey Fed with Silage Composed DietModified by Different Diet Anion and Cation Difference (DCAD)

The manuscript aimed to investigate the effect of corn silage with different values of dietary anion and cation (DCAD) supplements on growth rate and bone condition of turkeys. Six groups of turkey were used in this study; group A (control): standard diet (SD) (60%) 22 plus CS (40%); group B: SD (60%), CS (40%) plus 240g of CaCl2 per100kg of diet; group C: SD (60%), 23 CS (40%) plus 480g of CaCl2 per 100kg of diet; group D: SD (60%), CS (40%) plus 240g of NaHCO3 24 per 100kg of diet; group E: SD (60%), CS (40%) plus 480g NaHCO3 per 100kg of diet.  Individual body weight, DCAD values, and hydroxyproline concentration were measured.

The experimental hypothesis was that turkeys fed diets supplemented with corn sillage and either of CaCl2 or NaHCO3  will see a significant increase in body weight as well as improvement in bone condition. Although an interewting study, certain elements regarding experimental design and data interpretation reduce enthusiasm.

MAJOR CONCERNS

  • The Turkeys were fed in three phases; grower I, grower II, and finisher. What age did each feeding phase start and end? Only the phase duration was stated.
  • The analytical procedure described in line 101-106 is not clear. What is the rationale for these analyses?  What were the diets being analyzed for in this section?
  • On page 4, line 147, under control conditions in which the turkeys were only fed with standard diet and corn silage had the maximum value of DCAD. Why this?
  • The result section only showed values and comparisons was made without stating the significant difference values (p values, F values, DF). Also, the result stated that there was an increase in the body weight of turkey with both CaCl2 and NaHCO3, and increased hydroxyproline concentration whereas figure 1 and 2 show no significant differences.
  • Page 8, line 240-243, the sentence is incomplete.
  • Page 9, 258-259, the author stated that the DCAD values never resulted in pH lower than 7.2. Where are the values for the pH measured? A pH of 7.2 is very low physiologically.  The authors should have also monitored respiratory rates to determine if the birds were in metabolic acidosis and therefore under metabolic stress. 
  • The conclusion is not justified by the data. There are no mention in results or on figures of statistical differences in BW or hydroxyproline values.
  • The inference that changes in hydroxyproline directly relate to bone strength assumes quite a lot. The authors only did the hydroxyproline measures on one bone, the opposite bone should have been used for 3-point breaking test, or ashing, or other standard measures of bone strength in order to validate the hydroxyproline conclusions.  
  • In line 155, the authors state, “The addition of corn silage to the standard diet leads to a lower body weight in compari-155 son to the other groups.” However, according to their Methods EVERY group was given corn sileage thus no comparisons are possible, and this concoulsion is not appropriate.

MINOR CONCERNS

  • Page 1, line 23, 29 and 30, abbreviated words should be written in full.
  • Page 2, line 45, incomplete sentence.
  • Use of both indented and non-indented paragraphs.
  • Subtitles are written in lowercase instead of sentence case.
  • Uneven spacing between each subtitle.
  • Page 3, line 129, “famous” does not fit the context.
  • Incorrect reference citation.

Author Response

Thank you very much for your for critical but valuable comments. I tried to substantively refer to each comment. My answer are in red color.

Best regards
